# Age-Related Effects on MSC Immunomodulation, Macrophage Polarization, Apoptosis, and Bone Regeneration Correlate with IL-38 Expression

**DOI:** 10.3390/ijms25063252

**Published:** 2024-03-13

**Authors:** Jiewen Zhang, Kentaro Akiyama, Aung Ye Mun, Ryuji Tagashira, Tingling Zou, Naoya Matsunaga, Teisaku Kohno, Takuo Kuboki

**Affiliations:** Department of Oral Rehabilitation and Regenerative Medicine, Okayama University Graduate School of Medicine, Dentistry and Pharmaceutical Sciences, Okayama 700-8558, Japan; zhang-2019@s.okayama-u.ac.jp (J.Z.); aungyemun@s.okayama-u.ac.jp (A.Y.M.); ptal1cas@s.okayama-u.ac.jp (R.T.); pe592si4@s.okayama-u.ac.jp (T.Z.); kj73g89@s.okayama-u.ac.jp (N.M.); pr196bdl@s.okayama-u.ac.jp (T.K.); kuboki@md.okayama-u.ac.jp (T.K.)

**Keywords:** mesenchymal stem cell, aging, apoptosis, cytokines, monocytes and macrophages, immunomodulation

## Abstract

Mesenchymal stem cells (MSCs) are known to promote tissue regeneration and suppress excessive inflammation caused by infection or trauma. Reported evidence indicates that various factors influence the expression of MSCs’ endogenous immunomodulatory properties. However, the detailed interactions of MSCs with macrophages, which are key cells involved in tissue repair, and their regulatory mechanisms are not completely understood. We herein investigated how age-related immunomodulatory impairment of MSCs alters the interaction of MSCs with macrophages during bone healing using young (5-week old) and aged (50-week old) mice. To clarify the relationship between inflammatory macrophages (M1) and MSCs, their spatiotemporal localization at the bone healing site was investigated by immunostaining, and possible regulatory mechanisms were analyzed in vitro co-cultures. Histomorphometric analysis revealed an accumulation of M1 and a decrease in MSC number at the healing site in aged mice, which showed a delayed bone healing. In in vitro co-cultures, MSCs induced M1 apoptosis through cell-to-cell contact but suppressed the gene expression of pro-inflammatory cytokines by soluble factors secreted in the culture supernatant. Interestingly, interleukin 38 (*Il-38*) expression was up-regulated in M1 after co-culture with MSCs. IL-38 suppressed the gene expression of inflammatory cytokines in M1 and promoted the expression of genes associated with M1 polarization to anti-inflammatory macrophages (M2). IL-38 also had an inhibitory effect on M1 apoptosis. These results suggest that MSCs may induce M1 apoptosis, suppress inflammatory cytokine production by M1, and induce their polarization toward M2. Nevertheless, in aged conditions, the decreased number and immunomodulatory function of MSCs could be associated with a delayed M1 clearance (i.e., apoptosis and/or polarization) and consequent delayed resolution of the inflammatory phase. Furthermore, M1-derived IL-38 may be associated with immunoregulation in the tissue regeneration site.

## 1. Introduction

The skeletal system plays a crucial role in the human body and is one of the few tissues capable of self-repair [1]. However, in complex clinical situations, such as aging and degenerative diseases, like osteoporosis, or large bone defects due to trauma, tumor resection, or skeletal abnormalities, the self-repair ability of the skeletal system is significantly impaired [2]. Although skeletal-related diseases and conditions are rarely life threatening, they can dramatically impair function and reduce the quality of life [3]. 

Mesenchymal stem cells (MSCs) are multipotent stromal cells that exist in some adult tissues [4]. They are characterized by their ability to self-renew and differentiate into various cell types [5]. MSCs are potential candidates for tissue engineering and promote tissue regeneration by suppressing the excessive inflammation caused by infection or trauma [6] owing to their immunomodulatory functions. Previous reports have shown that systemic injection of MSCs can induce polarization of inflammatory macrophages (M1 polarized macrophages: M1) towards anti-inflammatory macrophages (M2 polarized macrophages: M2), improving the symptoms of immune dysregulation [7,8]. A number of studies have reported that the protective and regenerative capacities of MSCs decrease with age, leading to a reduction in the number and overall function of these cells. This results in increased inflammation and a slower healing process following injury [9]. 

In clinical practice, delayed wound healing associated with aging is a frequently encountered problem. Although it is well known that the balance between inflammation and regeneration is important in regulating tissue regeneration, the detailed mechanisms remain unclear. In particular, the mechanisms of immunomodulation at the site of tissue regeneration remain largely unknown.

To date, two main pathways have been elucidated for the immunomodulatory mechanisms of MSCs. One pathway involves the secretion of anti-inflammatory cytokines, which can suppress immune responses, such as the culture supernatant from MSCs can downregulate the expression pathways of tumor necrosis factor alpha (TNF-α) and interleukin 6 (IL-6) in macrophages by inhibiting the mitogen-activated protein kinase (MAPK) and nuclear factor kappa-B (NF-κB) [10,11]. The other pathway is based on direct cell-to-cell contact, such as the induction of T-cell apoptosis via the FAS and FAS Ligand (FASL) pathway, which may be a triggering factor for macrophage activation and regulatory T-cell differentiation [12]. Macrophages’ mass aggregation and proliferation can promote tissue regeneration at the initial stage of wound healing. But it may also lead to tissue necrosis if macrophage apoptosis is not properly coupled with phagocytic clearance [9]. However, some reports showed that the inflammation and oxidative stress-induced cellular senescence alter the immunomodulatory capacity of MSCs and impede their pro-regenerative function, leading to increased disease severity, maladaptive tissue injury, and development of complications [13,14,15]. Whether this age-related dysfunction of MSCs can affect tissue regeneration by altering MSCs’ interaction with macrophages, or more specifically altering the macrophage apoptosis, has not been clearly reported.

This study aimed to explore how age-related functional impairment affects the immunomodulatory properties of MSCs, more specifically, the interaction between MSCs and M1 macrophages, and their polarization to M2 phenotype. In more detail, we investigated the differences in the spatiotemporal localization of macrophages and MSCs and the distribution of apoptotic cells in a femoral bone defect model using young and aged mice. Additionally, detailed interactions were investigated by co-culturing MSCs with bone marrow-derived macrophages in vitro to study the induction of apoptosis in M1 and their polarity change to M2. 

## 2. Results

### 2.1. Delayed Femoral Bone Healing in Aged Mice

A femur bone defect model was developed in young (5-week-old) and aged (50-week-old) mice (Figure 1a,b). Micro-computed tomography (micro-CT) analysis revealed significantly more regenerated bone within the bone defect over time in the young groups than in the aged groups (*p* values for main effects of post-surgery time and mouse age and their interaction: 0.0002, 0.0003, and 0.0013, respectively; two-way factorial ANOVA), resulting in significant difference between the young and aged groups on day 7 (*p* value: 0.0002; Tukey’s post-hoc test) (Figure 1c and Appendix A). The regenerated bone area was also quantified based on hematoxylin and eosin (HE) stained areas that were widely expanded over time in the newly formed trabecular bone in the young mice, but not in the aged counterparts (*p* values for main effects of post-surgery time and mouse age and their interaction: 0.0006, 0.0007, and 0.0046, respectively; two-way factorial ANOVA) (Figure 1d and Appendix A). Similarly, Masson’s trichrome staining showed that the amount of aniline blue-stained mature bone tissue was significantly smaller in the aged group than in the young group (*p* values for main effects of post-surgery time and mouse age and their interaction: <0.0001, <0.0001, and <0.0001, respectively; two-way factorial ANOVA) (Figure 1e and Appendix A). Although the number of Runt-related transcription factor 2 positive (Runx2^+^) osteoblasts increased over time in the bone defect areas in both the young and aged groups (*p* values for main effects of post-surgery time and mouse age and their interaction: 0.0004, 0.0001, and 0.2689, respectively; two-way factorial ANOVA), it was significantly lower on day 3 in the aged group (*p* value: 0.0088; Tukey’s post-hoc test) (Figure 1f and Appendix A).

### 2.2. Delayed Macrophage Accumulation at the Wound Healing Site in Aged Mice

In the young group, the number of CD80^+^ M1 increased on day 3 and decreased on day 7 after surgery. In contrast, in the aged group, M1 number showed an increasing trend during the observation period (*p* values for main effects of post-surgery time and mouse age and their interaction: 0.0046, 0.0299, and 0.0020, respectively; two-way factorial ANOVA), resulting in significantly higher M1 number on day 7 (*p* value: 0.0047; Tukey’s post-hoc test) (Figure 2a and Appendix A). The number of CD206^+^ M2, which is known to be associated with the resolution of inflammation and tissue repair, also showed a time-dependent increase in both groups (*p* values for main effects of post-surgery time and mouse age and their interaction: 0.0003, 0.0032, and 0.5794, respectively; two-way factorial ANOVA). However, the aged group had a lower mean number than the young group at all time points (Figure 2b and Appendix A). 

### 2.3. Increased Number of Accumulated MSCs and Induce M1 Apoptosis in Young Mice

The percentage of platelet-derived growth factor receptor alpha positive (PDGFRα^+^) MSCs increased over time in both groups. Similar to M2, the number of MSCs was lower in aged mice than the young counterparts (*p* values for main effects of post-surgery time and mouse age and their interaction: 0.0002, 0.0052, and 0.8933, respectively; two-way factorial ANOVA) (Figure 3a and Appendix A). To detect apoptotic M1 macrophages, we performed double immunohistochemical staining for TdT-mediated UTP nick-end labeling (TUNEL) and CD80. The percentage of double-positive cells increased on day 3 after surgery (*p* values for main effects of post-surgery time and mouse age and their interaction: <0.0001, 0.0269, and 0.0009, respectively; two-way factorial ANOVA), and the number of apoptotic M1 cells in young mice was significantly higher than that in aged mice (*p* value: 0.0012; Tukey’s post-hoc test) (Figure 3b and Appendix A). 

### 2.4. Young MSCs Induced M1 Apoptosis through Cell-to-Cell Contact

Flow cytometric analysis showed that the mean percentage of apoptotic M1 was significantly increased after direct co-culture with MSCs but not with M0 and M2, while this result was partially abolished when M1 was co-cultured indirectly with MSCs using a cell culture insert (Transwell^®^) (Corning, NY, USA) (*p* values for main effects of macrophage types [M0/M1/M2] and experimental conditions [no MSCs/MSCs without Transwell/ MSCs with Transwell] and their interaction: <0.0001, <0.0001, and <0.0001, respectively; two-way factorial ANOVA) (*p* value for MSCs without Transwell vs. MSCs with Transwell: <0.0001; Tukey’s post-hoc test) (Figure 4a and Appendix A). Importantly, M1 co-cultured with aged MSCs showed a significantly lower mean percentage of apoptosis than those co-cultured with young MSCs (*p* values for main effects of macrophage types [M0/M1/M2] and MSC types [young/aged] and their interaction: <0.0001, 0.0153, and 0.0003, respectively; two-way factorial ANOVA) (Figure 4b and Appendix A). In support of these findings, the direct co-culture of young MSCs with M1 enhanced *Fas* expression in M1, while the direct co-culture of young MSCs with M2 did not enhance *Fas* expression in M2 (*p* values for main effects of macrophage types [M0/M1/M2] and experimental conditions [no MSCs/young MSCs/aged MSCs] and their interaction: <0.0001, <0.0001, and <0.0001, respectively; two-way factorial ANOVA) (Figure 4c and Appendix A). Regarding *Fasl* expression in MSCs, the direct co-culture of young MSCs with M1 enhanced the most apparent *Fasl* expression in MSCs (*p* values for main effects of experimental conditions [no macrophage/M0/M1/M2] and MSC types [young/aged] and their interaction: <0.0001, <0.0001, and 0.0003, respectively; two-way factorial ANOVA) (Figure 4c,d and Appendix A), resulting in significantly higher *Fasl* expression in MSCs co-cultured with M1 (*p* value for M1/young MSCs vs. M1/aged MSCs: <0.0001; Tukey’s post-hoc test) and with M2 (*p* value for M2/young MSCs vs. M2/aged MSCs: <0.0001; Tukey’s post-hoc test).

### 2.5. Young MSCs Inhibited Gene Expression of Inflammatory Cytokines of M1 through Soluble Factors

The analysis of gene expression using real-time reverse transcription–polymerase chain reaction (real-time RT-PCR) showed that the inflammatory genes, including *Il-6*, *iNos*, and *Tnf-α* expression in M1, were suppressed after indirect co-culture with young and aged MSCs. Note that this suppression was more evident when M1 was co-cultured with young MSCs (*p* values for main effect of experimental conditions [no MSC, + young MSCs, + aged MSCs] on *Il-6*, *iNos*, and *Tnf-α*: <0.0001, <0.0001, and <0.0001, respectively; one-way factorial ANOVA) (mean expression levels and *p* values with young vs. aged MSCs: *Il-6* [0.12 vs. 0.67, 0.0003], *iNos* [0.53 vs. 0.71, 0.0055], and *Tnf-α* [0.37 vs. 0.61, 0.0091]; Tukey’s post-hoc test) (Figure 5a and Appendix A). The gene expression of *Nf-κb*, a key intracellular regulator of inflammatory cytokine genes, was also significantly inhibited after M1 co-culture with young MSCs (*p* value for main effect of experimental conditions [M0, M1, M1+young MSCs, M1+aged MSCs]: 0.0008; one-way factorial ANOVA) (mean *Nf-κb* and *p* values with young vs. aged MSCs: 0.51 vs. 1.00, *p*=0.0297) (Figure 5b and Appendix A). Simultaneously, to assess M2 polarization, the expression of M2-related genes, such as *Arg-1*, *Ym-1*, and *Fizz-1*, was examined in M1 after co-culture with MSCs (*p* values for main effect of experimental conditions [no MSC, + young MSCs, + aged MSCs] on *Arg-1*, *Ym-1*, and *Fizz-1*: <0.0001, <0.0001, and <0.0001, respectively; one-way factorial ANOVA). The expression levels of these genes were significantly increased in M1 after co-culturing with young MSCs when compared to aged MSCs (mean expression levels and *p* values for young vs. aged MSCs: *Arg-1* [35.42 vs. 31.97, *p* = 0.3046], *Ym-1* [4.03 vs. 1.23, *p* = 0.0002], and *Fizz-1* [4.75 vs. 1.54, *p* < 0.0001]; Tukey’s post-hoc test) (Figure 5c and Appendix A). To further clarify whether these effects were caused by the activity of soluble factors derived from MSCs, culture supernatants collected after the co-culture of M1 and MSCs were added to another unstimulated M1 for culturing and analysis. Indeed, the supernatants from the co-cultures suppressed the gene expression of inflammatory cytokines and *Nf-κb* while enhancing the M2-related genes (*p* values for main effect of conditional medium types [M1, M1 + young MSCs, M1 + aged MSCs] on *Il-6*, *iNos*, *Tnf-α*, *Nf-κb*, *Arg-1*, *Ym-1*, and *Fizz-1*: 0.0056, 0.0207, 0.0014, 0.0042, <0.0001, <0.0001, and 0.0008, respectively; one-way factorial ANOVA). Interestingly, the suppressive effects on the gene expression of inflammatory cytokines and *Nf-κb* and the enhancing effects on the M2-related genes in M1 showed significant (*Il-6*, *iNos*, *Ym-1*, *Fizz-1*) or marginally significant (*Tnf-α*, *Nf-κb*) reduction when supernatants from aged MSCs were used (mean relative expression levels in M1 with young vs. aged MSCs and *p* values: *Il-6* [1.01 vs. 2.16, 0.0091], *iNos* [0.50 vs. 1.11, 0.0308], *Tnf-α* [0.57 vs. 0.76, 0.0466], *Nf-κb* [0.15 vs. 0.65, 0.0409], *Arg-1* [2.67 vs. 2.39, 0.1322], *Ym-1* [3.46 vs. 2.26, 0.0014], and *Fizz-1* [4.81 vs. 2.62, 0.0093]; Tukey’s post-hoc test) (Figure 5d–f and Appendix A). 

### 2.6. IL-38 Affected the Polarization of Macrophages and Apoptosis

Previous studies reported the anti-inflammatory effects of IL-38 [16], and its expression can induce macrophage activities in the efferocytosis process [17]. In this study, the gene expression level of *Il-38* in macrophages demonstrated a significant increase only when M1 and M2 were indirectly co-cultured with MSCs and the *Il-38* increase was more evident in M1 than in M2 (*p* values for main effects of macrophage types [M0/M1/M2] and experimental conditions [no MSCs/+ young MSCs/+ aged MSCs] and their interaction: <0.0001, <0.0001, <0.0001, respectively; two-way factorial ANOVA). In addition, this *Il-38* upregulation in M1 was significantly higher in co-cultures with young MSCs (mean relative expression levels of *Il-38* in M1 with young vs. aged MSCs and *p* values: 7.10 vs. 0.05, <0.0001; Tukey’s post-hoc test) (Figure 6a and Appendix A). For further confirmation, recombinant IL-38 (50 ng/mL) added into the M1 culture could suppress the gene expression of inflammatory cytokines and *Nf-κb* (*p* values for main effect of experimental conditions [M0, M1, M1 + IL-38] on *Il-6*, *iNos*, *Tnf-α*, and *Nf-κb*: <0.0001, <0.0001, <0.0001, 0.0283, respectively; one-way factorial ANOVA) (mean gene expression levels of the inflammatory cytokines and *Nf-κb* for M1 vs. M1 + IL-38 and *p* values: *Il-6* [73.55 vs. 59.09, 0.0278], *iNos* [4309 vs. 3429, 0.0118], *Tnf-α* [32.40 vs. 25.04, 0.0050], and *Nf-κb* [4.20 vs. 3.01, 0.2282]; Tukey’s post-hoc test) (Figure 6b,c and Appendix A). Meanwhile, IL-38 could increase the expression of anti-inflammatory M2-related genes (*p* values for main effect of experimental conditions [M0, M1, M1 + IL-38] on *Arg-1*, *Ym-1*, and *Fizz-1*: <0.0001, 0.0005, and <0.0001, respectively; one-way factorial ANOVA) (mean gene expression levels and *p* values for M1 vs. M1 + IL-38: *Arg-1* [0.80 vs. 10.72, <0.0001], *Ym-1* [0.46 vs. 1.31, 0.0004], and *Fizz-1* [0.55 vs. 1.01, <0.0001]; Tukey’s post-hoc test) (Figure 6d and Appendix A). Flow cytometric analysis revealed that the addition of IL-38 suppressed the apoptotic rate (%) in M1, which was significantly increased by co-culture with young MSCs (mean M1 apoptotic cell rates for M1 + young MSCs vs. M1 + young MSCs + IL-38 and *p* value: 19.53 vs. 8.97, <0.0001; Tukey’s post-hoc test). However, adding IL-38 did not change the percentage of apoptotic rate in M1 when co-cultured with aged MSCs (mean M1 apoptotic cell rates for M1 + aged MSCs vs. M1 + aged MSCs + IL-38 and *p* value: 10.97 vs. 10.60, 0.9985; Tukey’s post-hoc test) (Figure 6e and Appendix A). 

## 3. Discussion

Macrophages have been suggested to play essential roles in maintaining and repairing the tissue microenvironment, including pathogen elimination, immunosuppression, and anti-inflammatory responses. They may also be involved in inflammation-induced tissue regeneration [18,19,20,21,22]. On the other hand, MSCs are considered crucial cells for tissue regeneration due to their multipotent differentiation ability and immunomodulatory properties [6,22]. Recent studies have suggested that the crosstalk between these cells may be significant for tissue regeneration [23,24,25]. Pajarinen et al. showed that MSCs regulate the chemotaxis and function of macrophages, and that in some cases, the signaling derived from MSCs can promote bone regeneration by regulating macrophage function [24]. Ko et al. demonstrated that human MSCs can promote polarization of lung macrophages to M2 [26]. Our previous studies have shown that age-related dysfunction of MSCs is a key factor affecting the severity of periodontal tissue destruction severity in ligature-induced defects in mice [27]. The proliferation and differentiation ability of aged MSCs declines with changes in gene expression and cytokine production involved in tissue regeneration [28,29]. Duscher et al. have reported that aged MSCs fail to support the formation of vascular networks, thereby impeding their ability in age-related damage in cutaneous wound healing [30]. Studying the effects of age-related MSC dysfunction on the interaction between MSCs and macrophages during tissue regeneration and wound healing is very important to elucidate some of the mechanisms of age-related delayed healing that have been observed in clinical practice.

Our in vivo data, similar to previous reports [31,32], showed that M1 accumulated within 3 days postoperatively in young mice (Figure 2a). Subsequently, the number of MSCs and M2, which play an essential role in tissue regeneration, increased (Figure 2b and Figure 3a). This may indicate that accumulation of M1 in the early stages of inflammation may trigger a subsequent tissue regeneration cascade. Hence, we hypothesize that the delayed tissue regeneration in aged mice is accompanied by the delayed accumulation of M1. 

Numerous studies have reported that MSCs produce anti-inflammatory cytokines such as IL-4, IL-10, and IL-13 [33,34,35]. It is known that the production of these anti-inflammatory cytokines is regulated by various factors [15,25,36]. In the present study, the expression of anti-inflammatory cytokine genes such as *Tgf-β*, *Hgf*, and *Il-2* was increased with co-culture with inflammatory M1 (Appendix A), and significant changes in cytokine production were also observed in the comprehensive analysis (Appendix A). In addition, as IL-4 and IL-13 are well known as M2 inducers [37], gene expression levels of *Il-4* and *Il-13* in 50-week MSCs were significantly lower than that of 5-week MSCs after co-culture with M1 (Appendix A).

In terms of direct cell-to-cell contact, a prior study reported that systemically injected MSCs could induce T-cell apoptosis via the FAS/FASL pathway in both peripheral blood and bone marrow, leading to immunotolerance in systemic lupus erythematosus animals and humans [12]. However, it remains unclear whether accumulated endogenous MSCs induce immune cell apoptosis during local tissue regeneration. Our in vivo data revealed that with increasing numbers of accumulated MSCs, the number of apoptotic cells was increased on day 3 and then decreased on day 7 in young mice compared to that in aged mice (Figure 3b). In vitro confirmation demonstrated that MSCs from young mice promoted apoptosis at M1, whereas the induction effect was unclear at M0 and M2. M1 apoptosis induction was also observed in MSCs from aged mice but was significantly lower than that in young MSCs (Figure 4b). Moreover, this apoptosis-inducing effect was abolished by eliminating cell-to-cell contact from using the culture inserts (Figure 4a). Additionally, *Fas* expression in macrophages increased upon differentiation into M1, and *Fasl* expression was upregulated in young MSCs co-cultured with M1. In contrast, this effect was not observed in aged MSCs (Figure 4b,c). Macrophages in aged mice exhibited lower apoptosis when co-cultured with MSCs, despite *Fas* expression being upregulated in macrophages differentiated to M1 compared to macrophages from young mice (Appendix A). This suggests that age-related MSC dysfunction may be one of the factors affecting the induction of M1 apoptosis. Interestingly, this induction of apoptosis peaked at 3 days after bone defect creation and then decreased. There are several possible explanations for this phenomenon. One possible reason may be that M1 changed the polarity to M2 and no longer induced apoptosis, as shown by our in vitro co-culture results. Another possible reason could be a change in the induction of apoptosis in the accumulated cells. Among them, the apoptosis-inducing effect of accumulated MSCs, i.e., *Fasl* expression, may have been altered. In our previous study, *Fasl* expression of MSCs was increased by stimulation with inflammatory cytokines, and this effect was not significantly changed from 24 to 72 h after stimulation. In terms of this, it may be that the apoptosis-induced side, i.e., the apoptosis-induced M1, suppressed apoptosis induction as a negative feedback mechanism.

IL-38 is a new cytokine of the IL-1 family that shows 37–41% amino acid sequence homology with IL-1Ra and possesses anti-inflammatory properties [38,39,40]. Studies have reported that macrophages that release IL-38 regulate the cellular apoptotic balance and suppress inflammatory cytokine expression in macrophages [41] or Th17 [16]. Wei et al. reported that inflammatory cardiomyocytes release IL-38, suppressing excessive apoptosis and inflammation, and as a result, ventricular remodeling was alleviated [42]. However, the regulatory mechanism of IL-38 expression with its anti-inflammatory effect is not fully understood. Our results indicate that MSCs could promote its expression in M1, and this promoting function diminishes with age. Regarding the more specific anti-inflammatory effects of IL-38, we observed its suppression of gene expression of pro-inflammatory cytokines and *Nf-κb* in M1 (Figure 6b,c), consistent with previous reports [41]. Furthermore, in vitro results showed that IL-38 suppressed the onset of M1 inflammatory cytokines (Figure 6b,c) and enhanced the expression of M2-related genes (Figure 6d). Interestingly, IL-38 attenuated the apoptosis-inducing effect of cell-to-cell contact (Figure 6e). Although the expression of IL-38 at tissue regeneration sites and its relationship to the inhibition of M1 apoptosis is not clear, it will be considered as a possible negative feedback mechanism.

This study reveals that MSCs play an immunomodulatory function at the site of tissue regeneration by promoting the apoptosis of inflammatory macrophages, suppressing proinflammatory cytokines, and inducing polarity changes. The age-related decline in MSC function delays tissue regeneration and may play a part in the mechanism of delayed healing in elderly patients in clinic. However, the exact mechanism underlying IL-38 inhibition remains to be elucidated. Questions, such as how the delicate balance between MSC-induced M1 apoptosis and the anti-apoptotic effects of IL-38 is achieved, how MSC lose these functions with aging, and what can be done to activate the lost functions, need further investigation.

## 4. Materials and Methods

### 4.1. Animals

C57BL/6J mice (female, 5 and 50 weeks old) were purchased from Japan CLEA Co., Ltd. (Tokyo, Japan). The mouse femoral bone defect model was created under general anesthesia (ketamine, 90 mg/kg; hydroxyzine, 5 mg/kg). A central incision was made in the thigh skin to reveal the muscle. The muscle was then separated, and the periosteum was excised to expose the cortical bone surface of the femur. A bone defect was created in the femoral cortex, 3 mm above the knee joint in the 5-week-old group and 4 mm above in the 50-week-old group using a 1 mm diameter drill bit. All animals were maintained on a standard laboratory diet and were housed under the Okayama University Animal Research Committee (OKU-2021377) throughout the experimental period.

### 4.2. Micro-CT Analysis

Femurs were harvested 0, 3, or 7 days after surgery and fixed with 4% paraformaldehyde (PFA; Merck, Kenilworth, NJ, USA) for 24 h at 4 °C. The samples were then analyzed using micro-CT (SkyScan 1174, Bruker, Kontich, Belgium). The scanning protocol was a 6.5 μm voxel size, a peak voltage of 50 kV, 795 μA, and a 0.5 mm aluminum filter. Transmission images were reconstructed using SkyScan NReconc version 1.6.10.4 (Bruker, Germany). BV in the femur was analyzed at a volume of interest (VOI) with 0.5 mm side length square and 0.4 mm depth in the defect area by reconstructing the image using the SkyScan software (NReconc 1.6.10.4, CTAn 1.16.1.0, and Dataviewer 1.5.2.4 64-bit, SkyScan Germany). A single-blinded examiner reoriented the images and used the default threshold processor to perform measurements.

### 4.3. Histological Analysis

After fixation with 4% PFA, the femurs were decalcified with 10% ethylenediaminetetraacetic acid (EDTA) for 2–3 weeks and embedded in paraffin according to a standard protocol. For histological analysis, 7 μm thick sections were prepared and stained with HE and Masson’s trichrome. We utilized HE staining to color the samples and observed the tissue structure of the samples under a microscope for image analysis. For the measurement of bone reconstruction in Masson’s trichrome staining results, we followed the standard Masson’s trichrome staining procedure to highlight collagen fibers and bone tissue in blue. After HE and Masson’s trichrome staining, the samples were observed under a microscope, and ImageJ was used to quantify the area of newly formed bone tissue, thereby evaluating the extent of bone reconstruction.

### 4.4. Immunohistochemical Analysis

Non-fixed and undecalcified femurs were freeze-embedded in a super-cryo-embedding medium (SECTION-LAB Co., Ltd., Hiroshima, Japan), and 5 µm frozen sections were prepared with adhesive film according to the standard Kawamoto’s method [43]. The sections were then blocked with 5% goat serum (Life Technologies, Carlsbad, CA, USA) and stained with anti-CD80 monoclonal antibody (1:100, eBioscience, San Diego, CA, USA) for M1, anti-mouse MMR/CD206 antibody (1:50, R&D Systems, Minneapolis, MN, USA) for M2, anti-mouse PDGFRα antibody (1:100, R&D Systems, Minneapolis, MN, USA) for MSCs, and anti-Runx2 rabbit mAb (1:100, Cell Signaling, Danvers, MA, USA) for osteoblasts at 4 °C overnight. After washing, the specimens were incubated with secondary antibody Alexa Fluor 488, Goat anti-Hamster IgG (H+L) Cross-Absorbed Secondary Antibody (1:200, Life Technologies, Gaithersburg, MD, USA) for CD80 (M1), Alexa Fluor 594 chicken anti-Goat IgG (H+L) Cross-Adsorbed Secondary Antibody (1:200, Life Technologies, Carlsbad, CA, USA) for MMR/CD206 (M2), Alexa Fluor 594 chicken anti-goat IgG (1:200, Life Technologies, Carlsbad, CA, USA) for PDGFRα (MSCs), Alexa Fluor 594 goat anti-rabbit IgG (H+L) Cross-Adsorbed Secondary Antibody (1:200, Life Technologies, Carlsbad, CA, USA) for Runx2 (osteoblasts) for 60 min at room temperature. All images were captured using an all-in-one fluorescence microscope (BZ-710 Keyence, Osaka, Japan), and quantitative analyses of the number of positive cells per area in the regenerative healing site (ROI: 100 µm square) were performed using ImageJ software (ImageJ bundled with 64-bit Java 8, Bethesda, Rockville, MD, USA). 

### 4.5. TUNEL Assay

To obtain 5 µm frozen sections, the same methods that were used for immunohistochemical analysis were used for the TUNEL assay. According to the manufacturer’s instructions, Click-iT Plus TUNEL detection and Alexa Fluor 594 dye (Thermo Fisher Scientific, C10618, Waltham, MA, USA) vwere used to detect apoptotic cells in the bone defect area. Simultaneously, anti-CD80 monoclonal antibody and secondary antibody Alexa Fluor 488, Goat anti-Hamster IgG (H+L) Cross-Absorbed Secondary Antibody were used for CD80 (M1). All images were captured using an all-in-one fluorescence microscope (BZ-710 Keyence, Osaka, Japan), and quantitative analyses of the number of positive cells per area in the regenerative healing site (ROI: 100 µm square) were performed using ImageJ software (ImageJ bundled with 64-bit Java 8, Bethesda, Rockville, MD, USA).

### 4.6. Isolation of MSCs and Macrophages

The bone marrow from 5- or 50-week aged mice was flushed out with 2% Fetal Bovine Serum (FBS)/phosphate-buffered saline (PBS) medium through a 70 μm cell strainer (Greiner Bio-One Inc., Frickenhausen, Germany) to isolate MSCs or macrophages. Primary MSCs were seeded in Minimum Essential Medium Alpha (α-MEM, Life Technologies, Carlsbad, CA, USA) containing 15% FBS (Life Technologies, Carlsbad, CA, USA), 2 mM glutamate (Life Technologies, Carlsbad, CA, USA), 100 U/mL penicillin/streptomycin (Sigma-Aldrich, St Louis, MO, USA), and 55 μM 2-mercaptoethanol (2-ME, Life Technologies, Carlsbad, CA, USA) in 100 mm dishes (Greiner Bio-One Inc., Frickenhausen, Germany). For naïve macrophages (M0), the flushed bone marrow cells were cultured in Dulbecco’s modified MEM with high glucose (4.5 g/L), 10% FBS, 100 U/mL penicillin/streptomycin, and 100 ng/mL M-CSF (BioLegend, San Diego, CA, USA) in 6 well plates (Greiner Bio-One Inc., Frickenhausen, Germany). Non-adherent cells were eliminated by washing all dishes with PBS after 24 h. The colony-forming MSCs were picked and passaged for subculture after 7 days. The second passage of MSCs and naïve macrophages were used for further experiments.

### 4.7. In Vitro Co-Culture Experiments

The culture inserts (Transwell, Corning, Glendale, AZ, USA) for the 6-well plates were used for indirect co-culture. The naïve macrophages (M0) were seeded at a concentration of 1.0 × 10^6^ cells/well onto a 6-well plate (lower chamber) and induced to differentiate into M1 or M2 by adding100 ng/mL lipopolysaccharide (LPS, Sigma-Aldrich, Germany), 50 ng/mL IFN-γ (BioLegend, San Diego, CA, USA), or 20 ng/mL IL-4 (BioLegend, San Diego, CA, USA) for 24 h. After 24 h, culture dishes were washed with PBS and replaced with basal culture medium with 10%FBS. Then, MSCs (5- or 50-week MSCs were added into the culture insert (upper chamber) at a 1:1 ratio with the macrophages. In the control group, macrophages or MSCs were incubated separately in the same induction medium.

### 4.8. Real-Time RT-PCR

Total cellular RNA was extracted from MSCs and macrophages separately after 24 h of indirect co-culture using a Purelink RNA Mini Kit (Life Technologies, Carlsbad, CA, USA), according to the manufacturer’s instructions. Total RNA was reverse transcribed using the iScript cDNA Synthesis Kit (Bio-Rad, Hercules, CA, USA). Real-time RT-PCR was performed to quantify the expression of target genes using KAPA SYBR FAST qPCR Master Mix (KAPA BIOSYSTEMS, Wilmington, MA, USA) and a CFX96 real-time system (Bio-Rad, Hercules, CA, USA), as described previously [44]. The expression levels of each mRNA were normalized to those of the reference gene *s29*. The primer sequences are shown in Table 1.

### 4.9. Flow Cytometry Analysis

To separate MSCs and macrophages after direct co-culture, MSCs were pre-stained with PKH-26 (Invitrogen, Waltham, MA, USA), according to the manufacturer’s instructions. Apoptotic macrophages that were negative for PKH26 were detected using Annexin-V Apoptosis Detection Kit I (BD Pharmingen, Franklin Lakes, NJ, USA). All samples were analyzed using a BD Accuri C6 (BD Biosciences, Franklin Lakes, NJ, USA).

### 4.10. Statistical Analyses

The ROI was set as a square with a 0.7 mm side length for micro-CT and histological analysis and 0.1 mm side length for immunohistochemical analysis in the defected area with a 1 mm diameter and 0.8 mm depth from the cortical bone’s inner side.

Statistical analyses were performed by two-way factorial ANOVA with Tukey’s post-hoc correction test (Figure 1, Figure 2, Figure 3, Figure 4 and Figure 6a), or one-way factorial ANOVA with Tukey’s post-hoc correction test (Figure 5 and Figure 6), using GraphPad Prism version 9.5.1 for Windows (GraphPad Software, Boston, MA, USA) with a significance level set at *p* < 0.05. All data are presented as the mean ± standard deviation.

## Figures and Tables

**Figure 1 ijms-25-03252-f001:**
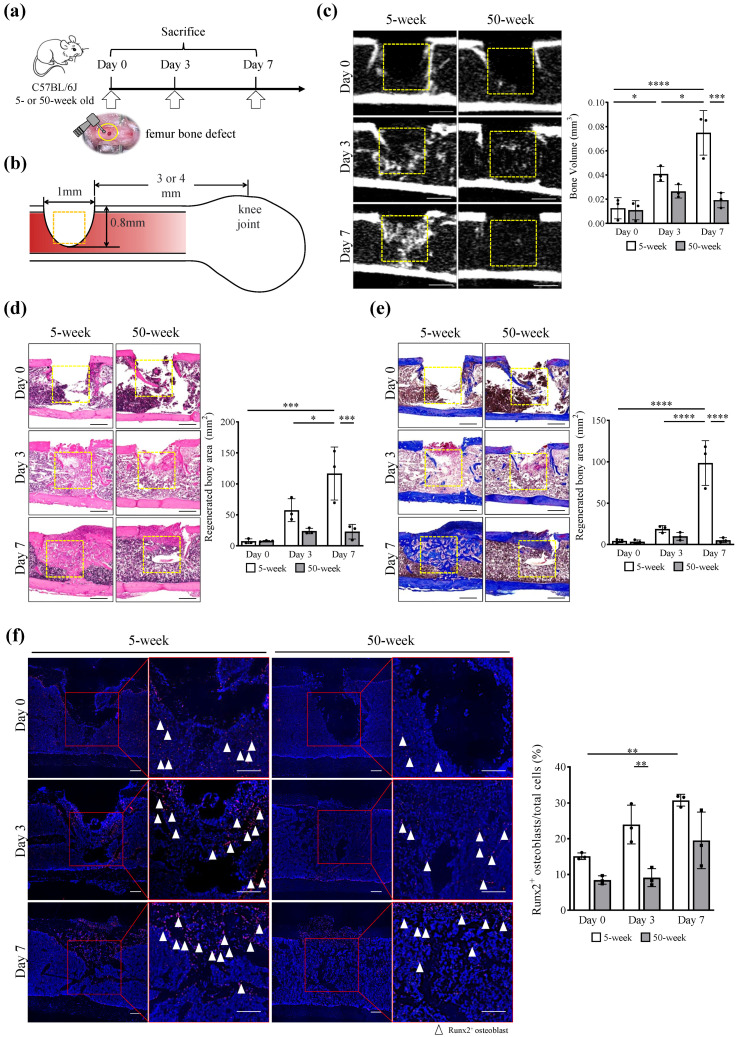
Aged mice showed delayed femoral bone healing. (**a**,**b**) Experimental design and establishment of the femoral bone defects in this study. Mice aged 5 and 50 weeks were used in the experiments. A bone defect was created using a 1 mm diameter round bar on the femoral cortex 3 or 4 mm above the knee joint. Mice were sacrificed at day 0, 3, or 7 after surgery. (**c**) Representative sagittal micro-CT images of the femoral defect after the surgery. Scale bar: 0.5 mm. Yellow square box: Region of Interest (ROI) used for quantification analysis. The bar graph shows the results of the quantitative analysis of bone volume (BV) of the defect area after surgery. (**d**) Histological examination by HE staining and (**e**) Masson’s trichrome staining after surgery. Scale bar: 0.5 mm. Yellow square box: ROI used for quantification. (**f**) Representative immunofluorescence images of Runx2^+^ osteoblasts (red) in the defect area after surgery. Cell nuclei were stained with DAPI (blue). Scale bar: 0.1 mm. Red square box: ROI used for quantification. Bone regeneration was measured by (**c**) BV, (**d**,**e**) regenerated bony area, and (**f**) Runx2^+^ osteoblast ratio in the defect area. Data are presented as mean ± standard deviation of three independent experiments or three mice. Two-way factorial ANOVA was performed to confirm statistical significance for main effects of both factors (post-surgery time and mouse age) and those interactions (Appendix A). Then, Tukey’s post-hoc tests were proceeded to confirm the difference between the bars (**** *p* < 0.0001, *** *p* < 0.001, ** *p* < 0.01, * *p* < 0.05, n = 3).

**Figure 2 ijms-25-03252-f002:**
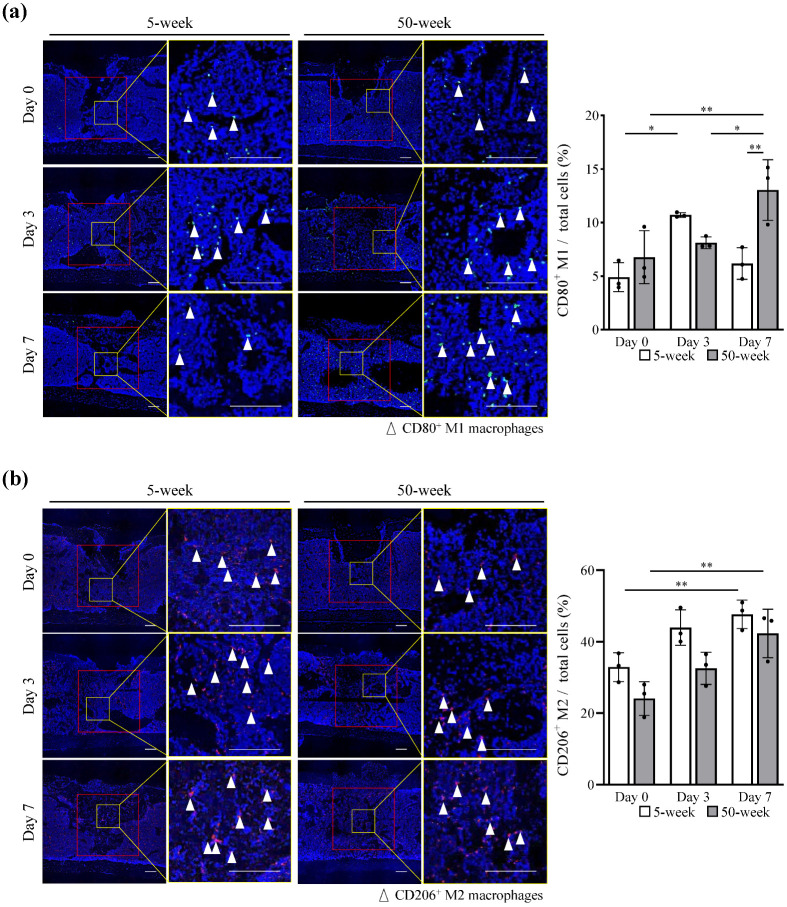
Aged mice showed decreased cell accumulation in the wound healing area. (**a**) Representative images of CD80^+^ inflammatory macrophages (M1) (green) and (**b**) CD206^+^ anti-inflammatory macrophages (M2) (red) in the defect area 0, 3, and 7 days after surgery. Cell nuclei were stained with DAPI (blue). Scale bar: 0.1 mm. Red square box: ROI used for quantification. Yellow square box: magnified images were shown on the right panel. The bar graph shows the results of the quantitative analysis of (**a**) CD80^+^ M1 or (**b**) CD206^+^ M2 in the ROI. Data are presented as the mean ± standard deviation of three independent experiments or three mice. Two-way factorial ANOVA was performed to confirm statistical significance for main effects of both factors (post-surgery time and mouse age) and those interactions (Appendix A). Then, Tukey’s post-hoc tests were proceeded to confirm the mean difference between the bars (** *p* < 0.01, * *p* < 0.05, n = 3).

**Figure 3 ijms-25-03252-f003:**
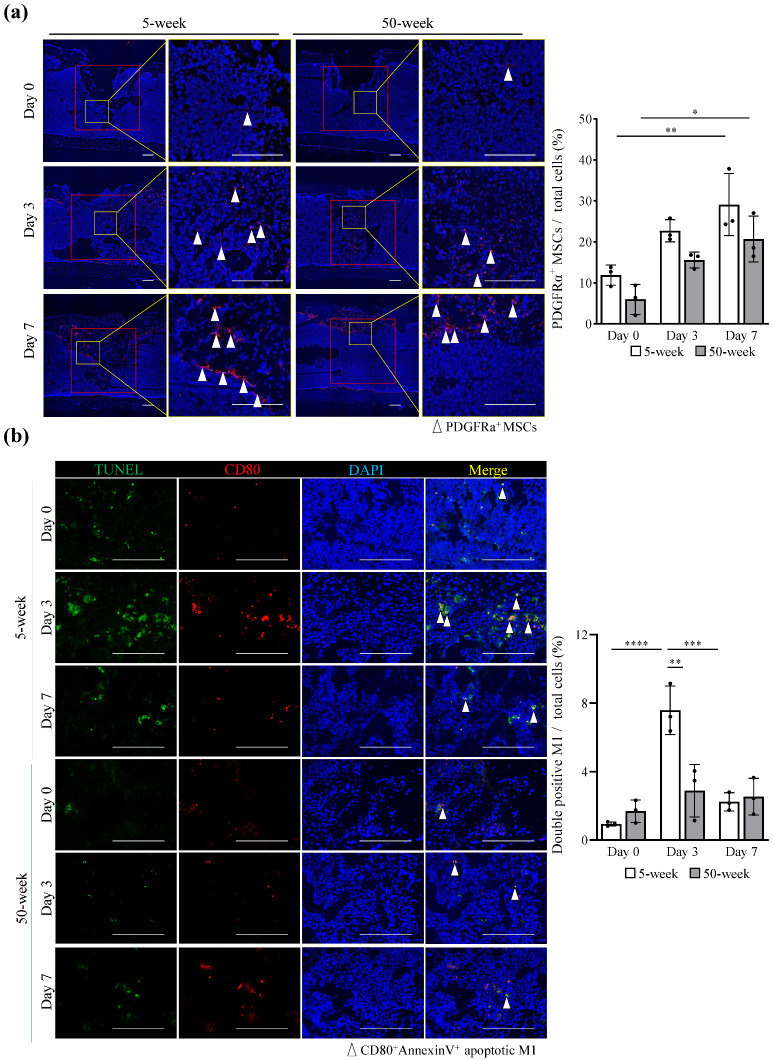
Increased accumulation of MSCs and induction of M1 apoptosis in young mice. (**a**) Representative images of PDGFRα^+^ MSCs (red) and (**b**) CD80^+^ M1 (green) and TUNEL^+^ (red) apoptotic M1 macrophages in the defect area 0, 3, and 7 days after surgery. Cell nuclei were stained with DAPI (blue). Scale bar: 0.1 mm. Red square box: ROI used for the quantification. Yellow square box: magnified images were shown on the right panel. The bar graph shows the results of quantitative analysis of (**a**) PDGFRα^+^ MSCs or (**b**) double-positive M1 in the ROI. Data are presented as the mean ± standard deviation of three independent experiments or three mice. Two-way factorial ANOVA was performed to confirm statistical significance for main effects of both factors (post-surgery time and mouse age) and those interaction (Appendix A). Then, Tukey’s post-hoc tests were proceeded to confirm the mean difference between the bars (**** *p* < 0.0001, *** *p* < 0.001, ** *p* < 0.01, * *p* < 0.05, n = 3).

**Figure 4 ijms-25-03252-f004:**
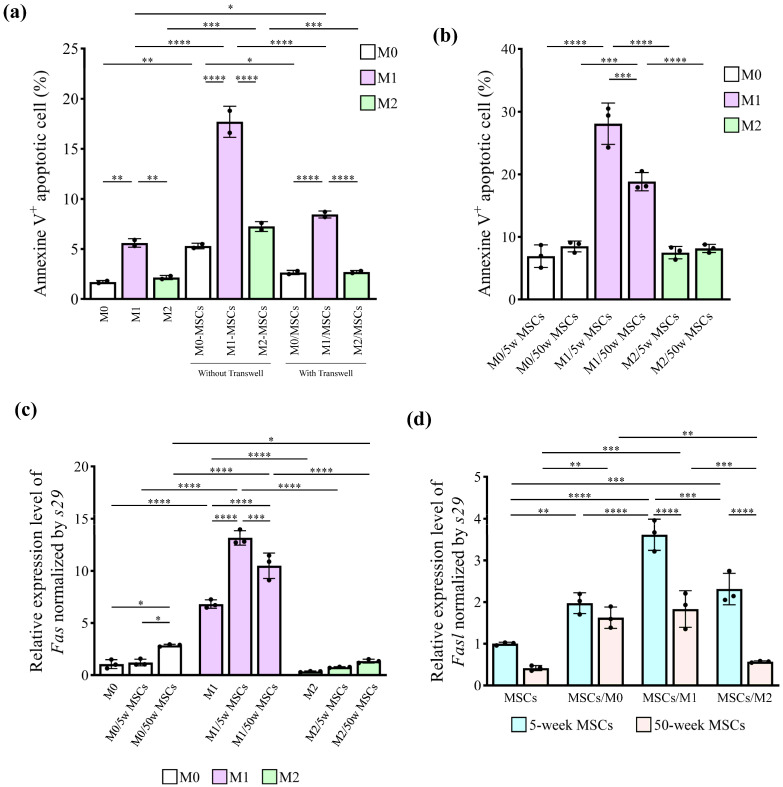
MSCs from aged mice showed low ability to induce M1 apoptosis via cell-to-cell contact. (**a**) Bone marrow-derived monocytes were stimulated with 100 ng/mL M-CSF for 6 h and then induced with 100 ng/mL lipopolysaccharide (LPS) and 50 ng/mL IFN-γ for M1, or 20 ng/mL IL-4 for M2 for 24 h. The 5-week MSCs were added to the Transwell (upper-chamber) at a 1:1 ratio with macrophages or without Transwell for 24 h, and the apoptotic cell percentage was measured by flow cytometry. (**b**) The same macrophages were cultured with 5- or 50-week MSCs directly at a 1:1 ratio, and the percentage of apoptotic cells was measured. (**c**) The bar graphs represent *Fas*’ relative gene expression levels in macrophages and (**d**) *Fasl* in 5- or 50-week MSCs after co-culture. Apoptotic cells were measured by flow cytometry (**a**,**b**), and gene expression levels were measured by real-time RT-PCR (**c**,**d**). Data are presented as the mean ± standard deviation of three independent experiments. Two-way factorial ANOVA was performed to confirm statistical significance for the main effects of both factors: (**a**) macrophage types and experimental conditions, (**b**) macrophage types and MSC types, (**c**) macrophage types and experimental conditions, (**d**) and MSC types and experimental conditions) and those interactions (Appendix A). Then, Tukey’s post-hoc tests were proceeded to confirm the mean difference between the bars (**** *p* < 0.0001, *** *p* < 0.001, ** *p* < 0.01, * *p* < 0.05, n = 3).

**Figure 5 ijms-25-03252-f005:**
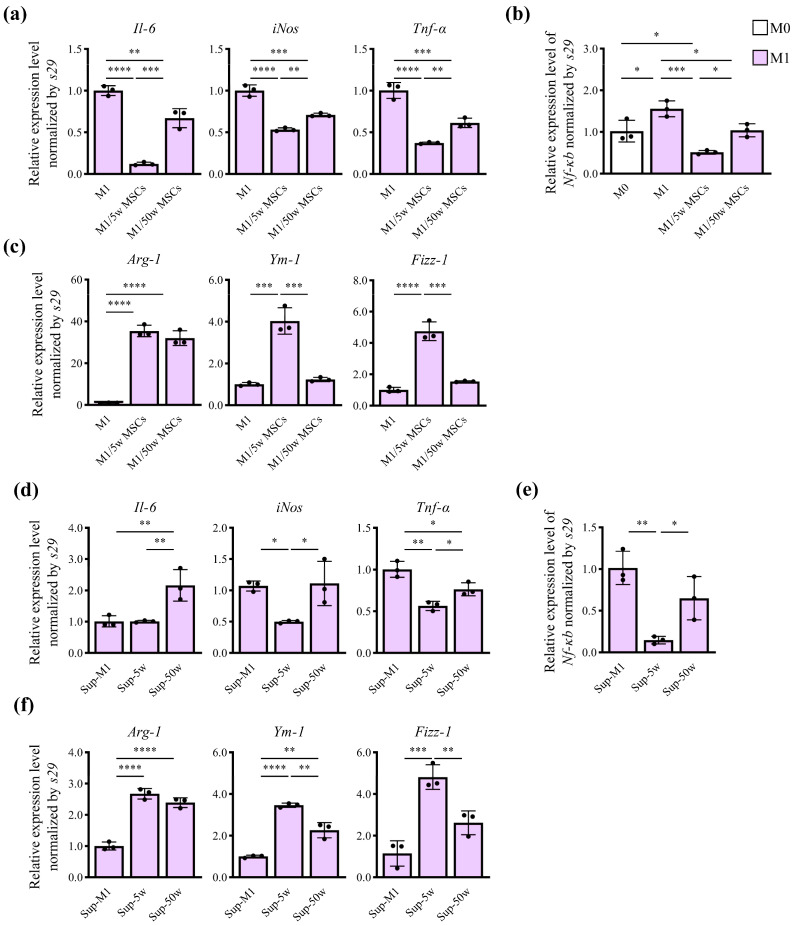
MSCs from aged mice showed low immunomodulatory properties through soluble factors. (**a**) Bone marrow-derived monocytes were stimulated with 100 ng/mL M-CSF for 6 h and then induced with 100 ng/mL LPS and 50 ng/mL IFN-γ to M1 for 24 h. The 5- or 50-week MSCs were added to the Transwell (upper-chamber) at a 1:1 ratio with macrophages for 24 h, and the relative gene expression levels of pro-inflammatory cytokines (*Il-6*, *iNos*, and *Tnf-α*), (**b**) *Nf-κb*, and (**c**) M2 macrophage-related genes (*Arg-1*, *Ym-1*, and *Fizz-1*) in M1 were measured by real-time RT-PCR. (**d**) The same macrophages were treated with the supernatant from M1 only, or M1 and MSCs co-culture, and then the relative gene expression levels of pro-inflammatory cytokines (*Il-6*, *iNos*, and *Tnf-α*), (**e**) *Nf-κb*, and (**f**) M2 macrophage-related genes (*Arg-1*, *Ym-1*, and *Fizz-1*) were investigated by real-time RT-PCR. Data are presented as mean ± standard deviation of three independent experiments. Data were analyzed using one-way factorial ANOVA with Tukey’s multiple comparisons test (Appendix A) (**** *p* < 0.0001, *** *p* < 0.001, ** *p* < 0.01, * *p* < 0.05, n = 3).

**Figure 6 ijms-25-03252-f006:**
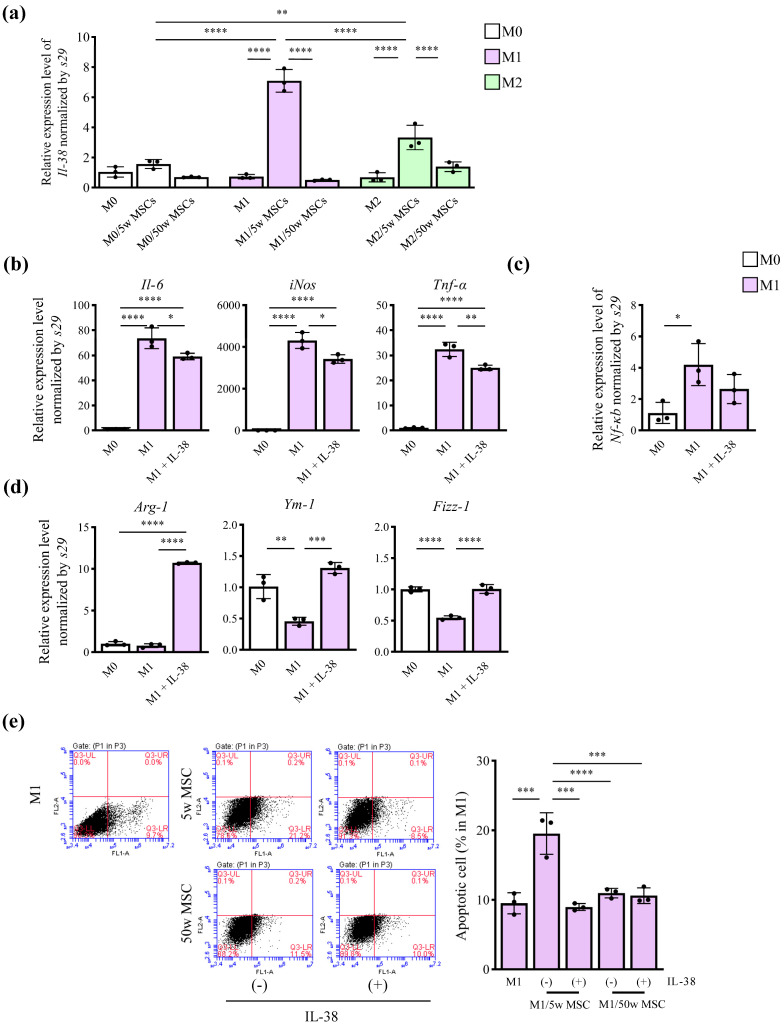
IL-38 plays an important role in macrophage polarization and apoptosis. (**a**) Bone marrow-derived monocytes were stimulated with 100 ng/mL M-CSF for 6 h and then induced with 100 ng/mL LPS and 50 ng/mL IFN-γ for M1 or 20 ng/mL IL-4 for M2 for 24 h; 5- or 50-week MSCs were added to the Transwell (upper chamber) at a 1:1 ratio with macrophages for 24 h, and the relative gene expression level of *Il-38* in macrophages was measured by real-time RT-PCR. (**b**) The same M1s were treated with IL-38 (50 ng/mL) for 24 h and the relative gene expression levels of pro-inflammatory cytokines (*Il-6*, *iNos*, and *Tnf-α*), (**c**) *Nf-κb*, and (**d**) M2 macrophage-related genes (*Arg-1*, *Ym-1*, and *Fizz-1*) were determined by real-time RT-PCR. (**e**) The percentage of apoptotic cells in M1 after co-culture with 5- or 50-week MSCs in the presence or absence of IL-38 (50 ng/mL) for 24 h was measured using flow cytometry. Data are presented as mean ± standard deviation of three independent experiments. (**a**) Two-way factorial ANOVA was performed to confirm the statistical significance for the main effects of both factors (macrophage types and experimental conditions) and those interaction (Appendix A). Then, Tukey’s post-hoc tests were proceeded to confirm the mean difference between the bars (**b**–**e**). Data were analyzed using one-way ANOVA with Tukey’s multiple comparison test (Appendix A) (**** *p* < 0.0001, *** *p* < 0.001, ** *p* < 0.01, * *p* < 0.05, n = 3).

**Table 1 ijms-25-03252-t001:** Nucleotide sequences of primers used for real-time RT-PCR.

Gene	Primer Sequences (5′-3′)	Product Size (bp)
** *s29* **	F, GGAGTCACCCACGGAAGTTCGG R, GGAAGCACTGGCGGCACATG	108
** *Fas* **	F, ATGCACACTCTGCGATGAAG R, CAGTGTTCACAGCCAGGAGA	120
** *Fas-l* **	F, GCAGAAGGAACTGGCAGAAC R, TTAAATGGGCCACACTCCTC	82
** *Il-6* **	F, TCCATCCAGTTGCCTTCT R, TAAGCCTCCGACTTGTGA	137
** *iNos* **	F, TGCATGGACCAGTATAAGGCAAGC R, GCTTCTGGTCGATGTCATGAGCAA	223
** *T* ** ** *nf-α* **	F, GTGGAACTGGCAGAACAG R, CACAAGCAGGAATGAGAAGA	96
** *Nf-κb* **	F, GGGGACTACGACCTGAATG R, GGGCACGATTGTCAAAGAT	118
** *Arg-1* **	F, ATGGAAGAGACCTTCAGCTAC R, GCTGTCTTCCCAAGAGTTGGG	224
** *Ym-1* **	F, GGGCATACCTTTATCCTGAG R, CCACTGAAGTCATCCATGTC	305
** *Fizz-1* **	F, TCCCAGTGAATACTGATGAGA R, CCACTCTGGATCTCCCAAGA	214
** *Il-38* **	F, GCCTGGCGTGTGTAAAGACA R, CCCTTGTATAGGTCCTCGATGTT	76
** *Tgf-β* **	F, CCCTATTTAAGAACACCCACTT R, GAGAAAGCAGCAGGAGTC	117
** *Hgf* **	F, TGGTTCTTGGTGTCATTGTT R, CCTCTTCTATGGCTATTACAACTT	137
** *Il-2* **	F, TTCTGAGGAGATGGATAGCC R, TGTGTTGTAAGCAGGAGGTA	81
** *Il-4* **	F, ACAGGAGAAGGGACGCCAT R, GAAGCCCTACAGACGAGCTCA	95
** *Il-13* **	F, AGACCAGACTCCCCTGTGCA R, TGGGTCCTGTAGATGGCATTG	123

## Data Availability

Data is contained within the article and Appendix A.

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
