# Peer review of "Age-Related Effects on MSC Immunomodulation, Macrophage Polarization, Apoptosis, and Bone Regeneration Correlate with IL-38 Expression"

_ijms, 2024, doi:10.3390/ijms25063252_

Round 1

Reviewer 1 Report

Comments and Suggestions for Authors

Dear authors,

Thank you for the opportunity to review your manuscript titled "The Age-related MSC Immunomodulatory Impairment Affects Macrophage Polarization and Apoptosis via IL-38, Resulting in Delayed Bone Regeneration." I have reviewed the manuscript, figures, and tables in detail. Here are my comments:

General comments:

  • The topic explored in this paper regarding the interaction between mesenchymal stem cells (MSCs) and macrophages during bone regeneration and how it is impacted by aging is novel and interesting. The experiments are well-designed to address the research aims.
  • The manuscript is generally well-written, but there are some areas needing improvement regarding language and grammar. I would recommend careful editing by a native English speaker.
  • The quality of the figures and tables is generally quite good. The images clearly demonstrate the results described.
  • The methods section provides adequate detail to understand and potentially reproduce the experiments performed.
  • The conclusion accurately summarizes the major findings of the study regarding the role of MSCs in modulating macrophage polarization and apoptosis during bone regeneration via IL-38 signaling.

Specific comments:

Introduction

  • Provide some more background on bone regeneration processes and the normal role of MSCs and macrophages. This will help readers better understand the context.
  • Expand a little on what is known about how aging impacts MSC function and bone regeneration.

Results

  • In Figure 1, please indicate in the legends the age of mice used in each group (young vs aged).
  • For Figures 2 and 3, it would be helpful to also show representative histology images along with the quantitative graphs.
  • In Figure 3b, the TUNEL assay specifically detects apoptotic cells. Please clarify in the text and figure legend that you are quantifying apoptotic M1 macrophages.
  • For Figure 4, consider including a schematic summarizing the key results regarding the differences in apoptotic effects observed with direct vs indirect MSC co-culture.

Discussion

  • Elaborate on the potential clinical implications of your findings related to age-related dysfunction of bone regeneration.
  • Discuss any limitations of the study as well as potential future research directions.

References

  • There are some inconsistencies in the formatting of the references. Please carefully check journal reference formatting guidelines.

In summary, this is an interesting and thoroughly executed study exploring the interaction between MSCs and macrophages during bone regeneration. With revisions addressing the comments raised regarding providing more context, improving clarity, and editing, I believe this manuscript could make a valuable contribution to the literature.

Best regards,

Comments on the Quality of English Language

Moderate changes required 

Author Response

Reviewer: 1

Thank you for the opportunity to review your manuscript titled "The Age-related MSC Immunomodulatory Impairment Affects Macrophage Polarization and Apoptosis via IL-38, Resulting in Delayed Bone Regeneration." I have reviewed the manuscript, figures, and tables in detail. Here are my comments:

General comments:

The topic explored in this paper regarding the interaction between mesenchymal stem cells (MSCs) and macrophages during bone regeneration and how it is impacted by aging is novel and interesting. The experiments are well-designed to address the research aims.

The manuscript is generally well-written, but there are some areas needing improvement regarding language and grammar. I would recommend careful editing by a native English speaker.

The quality of the figures and tables is generally quite good. The images clearly demonstrate the results described.

The methods section provides adequate detail to understand and potentially reproduce the experiments performed.

The conclusion accurately summarizes the major findings of the study regarding the role of MSCs in modulating macrophage polarization and apoptosis during bone regeneration via IL-38 signaling.

The authors thank the detailed comment addressed by the reviewer. We tried our best to check and polish grammar and language issues in this manuscript.

Specific comments:

Introduction

Provide some more background on bone regeneration processes and the normal role of MSCs and macrophages. This will help readers better understand the context.

Expand a little on what is known about how aging impacts MSC function and bone regeneration.

The authors thank the detailed comment addressed by the reviewer.  The additional information was described into introduction part (line: 64-67 and line: 85-87).

Results

In Figure 1, please indicate in the legends the age of mice used in each group (young vs aged).

The authors thank the detailed comment addressed by the reviewer.  The additional information was described into Figure 1.

For Figures 2 and 3, it would be helpful to also show representative histology images along with the quantitative graphs.

The authors thank the detailed comment addressed by the reviewer.  The representative histological images were provided into Figure 2 and 3.

In Figure 3b, the TUNEL assay specifically detects apoptotic cells. Please clarify in the text and figure legend that you are quantifying apoptotic M1 macrophages.

The authors thank the detailed comment addressed by the reviewer.  The additional descriptions were added into text (line 170) and figure legend (line 180).

For Figure 4, consider including a schematic summarizing the key results regarding the differences in apoptotic effects observed with direct vs indirect MSC co-culture.

The authors thank the detailed comment addressed by the reviewer.

The schematic summary of in vitro study was provided into Graphical abstract part.

Discussion

Elaborate on the potential clinical implications of your findings related to age-related dysfunction of bone regeneration.

Discuss any limitations of the study as well as potential future research directions.

The authors thank the detailed comment addressed by the reviewer.  The additional descriptions were added into discussion (line 386-394).

References

There are some inconsistencies in the formatting of the references. Please carefully check journal reference formatting guidelines.

The authors thank the detailed comment addressed by the reviewer. The modified reference according to the journal guidelines were provided in reference.

In summary, this is an interesting and thoroughly executed study exploring the interaction between MSCs and macrophages during bone regeneration. With revisions addressing the comments raised regarding providing more context, improving clarity, and editing, I believe this manuscript could make a valuable contribution to the literature.

The authors deeply thank the detailed and positive comments addressed by the reviewer.

Reviewer 2 Report

Comments and Suggestions for Authors

The manuscript describes experiments to determine if assumed age-related changes in MSCs alter the healing environment at sites of bone regeneration and differentially affect macrophage polarization. The data indicate that macrophage polarization is altered in young vs old mice at the bone defect site. The data also supports that MSCs from young vs old mice do differently affect macrophage polarization. These results support but do not prove a connection between aged MSCs and delayed bone healing. The justification for specifically calling out IL-38 for specific analysis should be better explained, perhaps some additional info in the introduction.

The manuscript is well organized but requires editing. The stats are generally confusing and the authors should consider ways to simplify the presentation and present the detailed stats elsewhere.

1.      What experiments were performed to insure that primary MSCs were in fact MSCs? Similarly, how were the M1 and M2 macrophages characterized?

2.      Group size for almost all experiments is 3 and the results are shown in bar graphs. The graphs should be changed to grouped scatter plots with bars and lines for means and SD.

3.      It is extremely difficult to see the described results in the immunofluorescence images provided when printed. Higher magnification and/or larger images are needed.

4.      The manuscript requires extensive editing.

5.      The animal model is somewhat problematic and requires some discussion regarding shortcomings. 3 mm from the knee in a 5 week old 10g mouse is not the same as 3 mm from the knee of a 40g mouse (50 wk). 3 mm would put the hole mid-diaphysis in a 5 wk mice but at the metaphysis in a 50 wk mouse. This point should be discussed.

6.      Title overstates the paper’s conclusions. There is no data showing that IL-38, for instance, reversed age-related delays in bone regeneration. Instead, consider: Age-related Effects on MSC Immunomodulation, Macrophage Polarization, Apoptosis, and Bone Regeneration correlate with IL-38 Expression

7.      The description of the methods used for microCT requires additional information such as an accurate description of voxel size and a description of thresholding procedures.

8.      Need to better define abbreviations. For instance, the 1st instance of “M1” occurs in the abstract and should be defined as M1 polarized macrophages.

9.      Figure 3B. Should show CD80 only data, TUNEL only data, and then double positive data.

10.  The description for culturing M0 macrophages is missing.

11.  For the co-culture experiments, when the MSCs were added (+/- transwell) was the media changed from the macrophages to media without LPS/IFNg or IL4 before adding MSCs?

12.  Figure 4A: Stats are not clear. Is there no difference between M1 and M2? This also holds for many other comparisons in other graphs.

13.  Figure 5 legend: panel e is missing; between “…and Tnf-a), NFkb, and (f)…..”

14.  Line 204: impress to induce?

15.  Figure 6. What is “NC”?

16.  Line 241: “guide” to “promote polarization of”

17.  Line 249: define “occurrence”

18.  Lines 260-262: This sentence is not clear.

19.  Line 314: “3 mm below” should be 3 mm above; “round bar” should this be a 1 mm dia. drill bit?

20.  Section 4.9: Were the TUNEL assays performed on frozen sections? What type of sample(s) were used for this assay?

21.  2D ROI vs 3D VOI: uCT vs histological analysis should be better defined throughout to more precisely define what volumes or areas were measured and analyzed.

22.  References are double-numbered.

23.  Figure S1: The source of the arrays should be stated and what is on the array should be described. Do the array dots correspond to the list of cytokines in the bar graph? If so, how do the two match up?

Comments on the Quality of English Language

Word choice and grammar need to be improved.

Author Response

Reviewer: 2

Comments and Suggestions for Authors

The manuscript describes experiments to determine if assumed age-related changes in MSCs alter the healing environment at sites of bone regeneration and differentially affect macrophage polarization. The data indicate that macrophage polarization is altered in young vs old mice at the bone defect site. The data also supports that MSCs from young vs old mice do differently affect macrophage polarization. These results support but do not prove a connection between aged MSCs and delayed bone healing. The justification for specifically calling out IL-38 for specific analysis should be better explained, perhaps some additional info in the introduction.

The manuscript is well organized but requires editing. The stats are generally confusing and the authors should consider ways to simplify the presentation and present the detailed stats elsewhere.

The authors deeply thank the detailed and positive comments addressed by the reviewer. 

  1. What experiments were performed to insure that primary MSCs were in fact MSCs? Similarly, how were the M1 and M2 macrophages characterized?

The authors thank the detailed comment addressed by the reviewer. To characterize the primary MSCs, the authors have already published detailed analysis in previous report (Aung et al., 2020, ref 27). Macrophages were also evaluated by checking the genes expression using real time PCR analysis.

  1. Group size for almost all experiments is 3 and the results are shown in bar graphs. The graphs should be changed to grouped scatter plots with bars and lines for means and SD.

The authors thank the detailed comment addressed by the reviewer.  The bar graphs were replaced as reviewer’s suggestion.

  1. It is extremely difficult to see the described results in the immunofluorescence images provided when printed. Higher magnification and/or larger images are needed.

The authors thank the detailed comment addressed by the reviewer.  The all pictures with higher resolution were replaced as reviewer’s suggestion.

  1. The manuscript requires extensive editing.

The authors thank the comment addressed by the reviewer.  The manuscript has been modified with English speaker.

  1. The animal model is somewhat problematic and requires some discussion regarding shortcomings. 3 mm from the knee in a 5 week old 10g mouse is not the same as 3 mm from the knee of a 40g mouse (50 wk). 3 mm would put the hole mid-diaphysis in a 5 wk mice but at the metaphysis in a 50 wk mouse. This point should be discussed.

The authors thank the detailed comment addressed by the reviewer. The additional explanation was added into materials and methods (line 406-407).

  1. Title overstates the paper’s conclusions. There is no data showing that IL-38, for instance, reversed age-related delays in bone regeneration. Instead, consider: Age-related Effects on MSC Immunomodulation, Macrophage Polarization, Apoptosis, and Bone Regeneration correlate with IL-38 Expression

The authors thank the detailed comment addressed by the reviewer. The revised title was provided as reviewers’ suggestion.

  1. The description of the methods used for microCT requires additional information such as an accurate description of voxel size and a description of thresholding procedures.

The authors thank the detailed comment addressed by the reviewer. The additional explanation was added into materials and methods (line 416-419).

  1. Need to better define abbreviations. For instance, the 1st instance of “M1” occurs in the abstract and should be defined as M1 polarized macrophages.

The authors thank the detailed comment addressed by the reviewer.

The explanations were modified in introduction (line 62-63) but not in abstract (because of word limitation).

  1. Figure 3B. Should show CD80 only data, TUNEL only data, and then double positive data.

The authors thank the detailed comment addressed by the reviewer.

The additional data were provided into Figure 3B.

  1. The description for culturing M0 macrophages is missing.

The authors thank the detailed comment addressed by the reviewer.

The additional information was provided into materials and methods (line 473).

  1. For the co-culture experiments, when the MSCs were added (+/- transwell) was the media changed from the macrophages to media without LPS/IFNg or IL4 before adding MSCs?

The authors thank the detailed comment addressed by the reviewer.

The additional information was provided into materials and methods (line 485-486).

  1. Figure 4A: Stats are not clear. Is there no difference between M1 and M2? This also holds for many other comparisons in other graphs.

The authors thank the detailed comment addressed by the reviewer.

The additional information was provided into all bar graphs.

  1. Figure 5 legend: panel e is missing; between “…and Tnf-a), NFkb, and (f)…..”

The authors thank the detailed comment addressed by the reviewer.

The additional description was added into figure legend (line 250).

  1. Line 204: impress to induce?

The authors thank the detailed comment addressed by the reviewer.

The sentence was corrected as reviewer’s’ suggestion (line 256).

  1. Figure 6. What is “NC”?

The authors thank the detailed comment addressed by the reviewer.

NC stand for “negative control” in this experiment. The description was modified as M0 in figure 6 b, c, and d.

  1. Line 241: “guide” to “promote polarization of”

The authors thank the detailed comment addressed by the reviewer.

The sentence was corrected as reviewer’s suggestion (line 298).

  1. Line 249: define “occurrence”

The authors thank the detailed comment addressed by the reviewer.

The sentence was modified (line 308).

  1. Lines 260-262: This sentence is not clear.

The authors thank the detailed comment addressed by the reviewer.

The sentence was modified (line 300-304).

  1. Line 314: “3 mm below” should be 3 mm above; “round bar” should this be a 1 mm dia. drill bit?

The authors thank the detailed comment addressed by the reviewer.

The sentence was corrected as reviewer’s suggestion (line 406-407).

  1. Section 4.9: Were the TUNEL assays performed on frozen sections? What type of sample(s) were used for this assay?

The authors thank the detailed comment addressed by the reviewer.

The additional information was provided into materials and methods (line 455-463).

  1. 2D ROI vs 3D VOI: uCT vs histological analysis should be better defined throughout to more precisely define what volumes or areas were measured and analyzed.

The authors thank the detailed comment addressed by the reviewer.

The additional information was provided into materials and methods (line 416-419).

  1. References are double-numbered.

The authors thank the detailed comment addressed by the reviewer.

The references were corrected as reviewer’s suggestion.

  1. Figure S1: The source of the arrays should be stated and what is on the array should be described. Do the array dots correspond to the list of cytokines in the bar graph? If so, how do the two match up?

The authors thank the detailed comment addressed by the reviewer.

According to the manufacturer's instructions, the mouse cytokine array panel A was used for the supernatants from co-culture systems. The array coordinate is followed.

Comments on the Quality of English Language

Word choice and grammar need to be improved.

The authors thank the comment addressed by the reviewer.  The manuscript has been modified with English speaker.

Reviewer 3 Report

Comments and Suggestions for Authors

Zhang et al. investigated the relationship between M1 and MSCs in a tissue regeneration process in aged animal using a mouse bone-defect model. The topic is interesting, however, throughout the manuscript there are many conclusive statements not sufficiently supported by experimental evidences. Furthermore, often vague expressions and grammatical errors hampers for readers to properly understand the results and authors´ statements. The figures omits some critical information and was not properly organized leading to great confusion for readers. I believe all the figures have to be largely reorganized. The critical weaknesses as below should be reconsidered before submission.

1. Introduction:  As authors also pointed out the importance of age-related dysfunction of MSCs for issue of the present study, authors should properly introduce the published characteristics of age-related dysfunction of MSCs. These literature-based knowledge may lead the context of the present study to be more comprehensive. These introductions also make reader to understand what the present study aims for. For example, why the study was specifically interested in M1 to M2 transition, and which age-related dysfunction of MSC are critically important for tissue regeneration, etc.

2. The numbers of animal or specimens should be specified in each data presented in figures, results and material and methods section. In histomorphometric analysis, it is also critical. In a method section, the methods of the histomorphometric analysis should be described in detail.

3. All figures should be re-organized.

Figure1-3: it would be better to see if target cell or protein stained would be shown in the left panels as immunostaining figures.

The tables  for Time/age variables (Fig. 1-3) should be placed as separate tables for example, in a suppl. Table, because at present figures it is very confusing with presenting variables in graphs.

Figure 4 c,d): what is (-)?

Figure 4-6: I am totally confused with the presented data, because all variables and parameters are all different each figures and graphs.

For example, MSC(-), MSC, (-), MSC(+), sometimes M0 sometimes without M0…….

The data presented makes me really difficult to interpret the data properly. It has to be completely re-organized as a unified form as possible.

4. Grammatical errors and unclear statements:

Abstract: M1? M2?

Line 28: “which might be a starting point for tissue regeneration at the wound healing sites.” (?)  

Discussion: Line 244: “..studying age-related dysfunction of bone marrow-derived MSCs affect the interaction of MSCs and macrophages during tissue regeneration and wound healing processes would be of great value.” (?)

Line 249: “..after occurrence..” (?)

Line 250: “.. , and the accumulation of MSCs, which conduct immunotolerance in local inflammatory sites, time,…” (?)

Line 258, 268, 275: “increased” (?) : might be ´ was increased´ (?)

Line 260: “Notably, Il-4 and Il-13 also play as the M2-induced gene during the immunomodulatory, the gene expression levels were markedly reduced in 50-week MSCs after co-culture with M1” (?)

Line 271: apoptosis induction (?)

 5. Many conclusive statements not sufficiently supported by experimental evidences.

Line 253: “Therefore, tissue regeneration may be delayed in old mice, with delayed M1 accumulation. “   

Line 280: “This suggests that the induction of M1 apoptosis may depend on the functional impairment of MSCs.”

Line 283: “several possible explanations” : it should be fully discussed not by short sentences only for only 2 possible reasons.

Line 296, 300: “IL-38 polarity-inducing effect” “an anti-inflammatory effect” “negative feedback regulatory effect” (?)

Regarding IL-38, additional experiments should be added if authors would propose as described in the discussion and abstract. In the discussion, It should be further discussed about IL-38 polarity-inducing effect, an anti-inflammatory effect and negative feedback regulatory effect with previous studies in literature if authors would intend IL-38 to place as a critical role in the relationship between M1 and MSCs in a tissue regeneration process in aged animal.

Overall, the manuscript has to be largely re-written with additional experiments before submitting the manuscript.

Comments on the Quality of English Language

I believe extensive editing of English language required.

Author Response

Reviewer: 3

Zhang et al. investigated the relationship between M1 and MSCs in a tissue regeneration process in aged animal using a mouse bone-defect model. The topic is interesting, however, throughout the manuscript there are many conclusive statements not sufficiently supported by experimental evidences. Furthermore, often vague expressions and grammatical errors hampers for readers to properly understand the results and authors´ statements. The figures omits some critical information and was not properly organized leading to great confusion for readers. I believe all the figures have to be largely reorganized. The critical weaknesses as below should be reconsidered before submission.

 The authors deeply thank the detailed and positive comments addressed by the reviewer. 

  1. Introduction:  As authors also pointed out the importance of age-related dysfunction of MSCs for issue of the present study, authors should properly introduce the published characteristics of age-related dysfunction of MSCs. These literature-based knowledge may lead the context of the present study to be more comprehensive. These introductions also make reader to understand what the present study aims for. For example, why the study was specifically interested in M1 to M2 transition, and which age-related dysfunction of MSC are critically important for tissue regeneration, etc.

The authors thank the detailed comment addressed by the reviewer.  The additional description was provided into introduction part (line: 85-88).

  1. The numbers of animal or specimens should be specified in each data presented in figures, results and material and methods section. In histomorphometric analysis, it is also critical. In a method section, the methods of the histomorphometric analysis should be described in detail.

The authors thank the detailed comment addressed by the reviewer.  The additional description was provided into materials and methods part (line: 426-434).

  1. All figures should be re-organized.

Figure1-3: it would be better to see if target cell or protein stained would be shown in the left panels as immunostaining figures.

The authors thank the detailed comment addressed by the reviewer.

The modified figures were provided in Figure 1-3.

The tables for Time/age variables (Fig. 1-3) should be placed as separate tables for example, in a suppl. Table, because at present figures it is very confusing with presenting variables in graphs.

The authors thank the detailed comment addressed by the reviewer.

Figure 4 c,d): what is (-)?

The authors thank the detailed comment addressed by the reviewer.

The authors used (-) as without MSCs or transwell. Now the descriptions have been modified as Figure 4 a, b, c.

Figure 4-6: I am totally confused with the presented data, because all variables and parameters are all different each figure and graphs.

For example, MSC (-), MSC, (-), MSC (+), sometimes M0 sometimes without M0…….

The data presented makes me really difficult to interpret the data properly. It has to be completely re-organized as a unified form as possible.

The authors thank the detailed comment addressed by the reviewer.

The authors modified all figures as reviewer’s suggestion.

  1. Grammatical errors and unclear statements:

Abstract: M1? M2?

The authors thank the detailed comment addressed by the reviewer.

The additional descriptions were added into introduction part. (line: 28 and 35)

Line 28: “which might be a starting point for tissue regeneration at the wound healing sites.” (?)  

The authors thank the detailed comment addressed by the reviewer.

Exaggerated sentences have been corrected. (line: 35-41)

Discussion: Line 244: “..studying age-related dysfunction of bone marrow-derived MSCs affect the interaction of MSCs and macrophages during tissue regeneration and wound healing processes would be of great value.” (?)

The authors thank the detailed comment addressed by the reviewer.

Exaggerated sentences have been corrected. (line: 300-304)

Line 249: “..after occurrence..” (?)

The authors thank the detailed comment addressed by the reviewer.

The sentence was corrected. (line: 308)

Line 250: “.. , and the accumulation of MSCs, which conduct immunotolerance in local inflammatory sites, time,…” (?)

The authors thank the detailed comment addressed by the reviewer.

Exaggerated sentences have been corrected. (line: 307-312)

Line 258, 268, 275: “increased” (?) : might be ´ was increased´ (?)

The authors thank the detailed comment addressed by the reviewer.

The sentence was corrected. (line: 323, 336, 357)

Line 260: “Notably, Il-4 and Il-13 also play as the M2-induced gene during the immunomodulatory, the gene expression levels were markedly reduced in 50-week MSCs after co-culture with M1” (?)

The authors thank the detailed comment addressed by the reviewer.

Exaggerated sentences have been corrected. (line: 325-328)

Line 271: apoptosis induction (?)

The authors thank the detailed comment addressed by the reviewer.

The sentence was corrected. (line: 339)

  1. Many conclusive statements not sufficiently supported by experimental evidences.

Line 253: “Therefore, tissue regeneration may be delayed in old mice, with delayed M1 accumulation. “   

The authors thank the detailed comment addressed by the reviewer.

Exaggerated sentences have been corrected. (line: 307-312)

Line 280: “This suggests that the induction of M1 apoptosis may depend on the functional impairment of MSCs.”

The authors thank the detailed comment addressed by the reviewer.

Exaggerated sentences have been corrected. (line: 348-349)

Line 283: “several possible explanations” : it should be fully discussed not by short sentences only for only 2 possible reasons.

The authors thank the detailed comment addressed by the reviewer.

The additional sentences were added into discussion part (line 352-360).

Line 296, 300: “IL-38 polarity-inducing effect” “an anti-inflammatory effect” “negative feedback regulatory effect” (?)

Regarding IL-38, additional experiments should be added if authors would propose as described in the discussion and abstract. In the discussion, It should be further discussed about IL-38 polarity-inducing effect, an anti-inflammatory effect and negative feedback regulatory effect with previous studies in literature if authors would intend IL-38 to place as a critical role in the relationship between M1 and MSCs in a tissue regeneration process in aged animal.

The authors thank the detailed comment addressed by the reviewer.

Exaggerated sentences have been corrected. (line373-378)

Overall, the manuscript has to be largely re-written with additional experiments before submitting the manuscript.

Comments on the Quality of English Language

I believe extensive editing of English language required

The authors thank the comment addressed by the reviewer.  The manuscript has been modified with English speaker.

Round 2

Reviewer 1 Report

Comments and Suggestions for Authors

Dear authors,

Thank you for the opportunity to re-review your revised manuscript titled "Age-related Effects on MSC Immunomodulation, Macrophage Polarization, Apoptosis, and Bone Regeneration correlate with IL-38 Expression." I appreciate you addressing the concerns raised in my initial review. The manuscript has improved, and most of my comments have been adequately addressed. Please find some remaining suggestions below:

General comments:

  • The overall quality of the writing, figures, methods, and presentation of results is high. The topic is interesting and important.
  • The revisions have enhanced the impact and clarity of the study. The expanded introduction and discussion, in particular, provide helpful additional context.

Specific comments:

Introduction

  • Well improved with more background on macrophages, MSCs, and impacts of aging. This section now effectively sets the stage.

Results

  • Figures are clear, adequately referenced in the text, and reinforce the major findings. No further suggestions.

Discussion

  • You have expanded nicely on implications and limitations. One very minor additional limitation to consider would be the focus only on bone regeneration here; effects may differ in other tissues/organs.

References

  • References are now appropriately formatted and cited. Self-citations seem reasonable. No concerns.

In summary, I am now supportive of publication after the minor remaining edits. The topic is intriguing, the methods are sound, the data is convincing, and the writing explains the story well. Congratulations on developing an important contribution to further understanding immunomodulation in tissue repair. I wish you the very best as you continue moving forward with this work.

Comments on the Quality of English Language

Minor editing of English language required

Author Response

Reviewer(s)’ Comments to Author:

Reviewer: 1 (Round 2)

Thank you for the opportunity to re-review your revised manuscript titled "Age-related Effects on MSC Immunomodulation, Macrophage Polarization, Apoptosis, and Bone Regeneration correlate with IL-38 Expression." I appreciate you addressing the concerns raised in my initial review. The manuscript has improved, and most of my comments have been adequately addressed. Please find some remaining suggestions below:

General comments:

The overall quality of the writing, figures, methods, and presentation of results is high. The topic is interesting and important.

The revisions have enhanced the impact and clarity of the study. The expanded introduction and discussion, in particular, provide helpful additional context.

The authors thank the positive comment addressed by the reviewer.

Specific comments:

Introduction

Well improved with more background on macrophages, MSCs, and impacts of aging. This section now effectively sets the stage.

Results

Figures are clear, adequately referenced in the text, and reinforce the major findings. No further suggestions.

Discussion

You have expanded nicely on implications and limitations. One very minor additional limitation to consider would be the focus only on bone regeneration here; effects may differ in other tissues/organs.

The authors thank the detailed comment addressed by the reviewer.

The additional information was described into introduction part.

References

References are now appropriately formatted and cited. Self-citations seem reasonable. No concerns.

In summary, I am now supportive of publication after the minor remaining edits. The topic is intriguing, the methods are sound, the data is convincing, and the writing explains the story well. Congratulations on developing an important contribution to further understanding immunomodulation in tissue repair. I wish you the very best as you continue moving forward with this work.

The authors deeply thank the detailed and positive comments addressed by the reviewer.

Reviewer 2 Report

Comments and Suggestions for Authors

The authors have significantly revised the manuscript and satisfactorily addressed many of the prior concerns.

Concerns remaining include:

1.      Difficulty seeing the immunofluorescence (IF) in many of the IF figures. Even when the figures are enlarged in the manuscript pdf, the results are often (not always) difficult to see.

2.      M1 is still not defined at 1st instance in abstract, but is instead defined at the second instance in the abstract. This should be corrected.

3.      microCT parameters are still not properly defined. Voxel size should be described in microns, not pixels, and a voxel defines a volume, not a linear distance.

4.      There are numerous spelling and syntax errors that will need to be corrected.

Most importantly, the authors statistical analysis is  problematic. For instance, it is not appropriate to do a 3-way ANOVA, determine that a parameter is significant, and then perform a 2-way ANOVA using that parameter. Post-hoc testing should be performed as part of the 3-way ANOVA so that any corrections (Tukey, as used here) will be appropriately applied. It is strongly recommended that a statistician review the manuscript and correct the stats

Comments on the Quality of English Language

Minor editing is required.

Author Response

Reviewer: 2 (Round 2)

Comments and Suggestions for Authors

The authors have significantly revised the manuscript and satisfactorily addressed many of the prior concerns.

Concerns remaining include:

  1. Difficulty seeing the immunofluorescence (IF) in many of the IF figures. Even when the figures are enlarged in the manuscript pdf, the results are often (not always) difficult to see.

The authors thank the detailed comment addressed by the reviewer.

The original PPT file was submitted for publication.

  1. M1 is still not defined at 1st instance in abstract, but is instead defined at the second instance in the abstract. This should be corrected.

The authors thank the detailed comment addressed by the reviewer.

The explanations were modified in abstract part.

  1. microCT parameters are still not properly defined. Voxel size should be described in microns, not pixels, and a voxel defines a volume, not a linear distance.

The authors thank the detailed comment addressed by the reviewer.

The additional explanation was added into materials and methods part.

  1. There are numerous spelling and syntax errors that will need to be corrected.

The authors thank the detailed comment addressed by the reviewer.

The spelling and syntax errors were corrected in the manuscript.

Most importantly, the authors statistical analysis is problematic. For instance, it is not appropriate to do a 3-way ANOVA, determine that a parameter is significant, and then perform a 2-way ANOVA using that parameter. Post-hoc testing should be performed as part of the 3-way ANOVA so that any corrections (Tukey, as used here) will be appropriately applied. It is strongly recommended that a statistician review the manuscript and correct the stats

The authors thank the detailed comment addressed by the reviewer.

The all-data analysis was performed again and add corrections in the Results part.